# Association of tuberculosis risk with genetic polymorphisms of the immune checkpoint genes *PDCD1*, *CTLA-4*, and *TIM3*

Chi-Wei Liu[1,2☯], Lawrence Shih-Hsin Wu[3,4☯], Chou-Jui Lin[1], Hsing-Chu Wu[1], Kuei-Chi Liu[1], Shih-Wei Lee[1,5]*

**1** Department of Internal Medicine, Taoyuan General Hospital, Ministry of Health and Welfare, Taoyuan, Taiwan, **2** Translational Medicine Center, Taoyuan General Hospital, Department of Health and Welfare, Taoyuan, Taiwan, **3** Graduate Institute of Biomedical Sciences, China Medical University, Taichung, Taiwan, **4** Center for Allergy, Immunology, and Microbiome (A.I.M.), China Medical University Hospital, Taichung, Taiwan, **5** Department of Nursing, Yuanpei University of Medical Technology, Hsinchu, Taiwan

☯ These authors contributed equally to this work.

* chestman9@gmail.com

**Data Availability Statement:** All data generated or analyzed during this study are included in this published article and its supplementary information files.

## Abstract

The immune checkpoint proteins were reported to involve to host resistance to *Mycobacteria tuberculosis* (*Mtb*). Here, we evaluated 11 single nucleotide polymorphisms (SNPs) in *PDCD1*, *CTLA4*, and *HAVCR2* genes between participants with and without TB infection. Genomic DNA isolated from 285 patients with TB and 270 controls without TB infection were used to perform the genotyping assay. Odds ratios were used to characterize the association of 11 SNPs with TB risk. In this study, the various genotypes of the 11 SNPs did not differ significantly in frequency between the non-TB and TB groups. When patients were stratified by sex, however, men differed significantly from women in genotype frequencies at *HAVCR2* rs13170556. Odds ratios indicated that rs2227982, rs13170556, rs231775, and rs231779 were sex-specifically associated with TB risk. In addition, the combinations of rs2227982/rs13170556 GA/TC in men and the A-C-C haplotype of rs231775-rs231777-rs231779 in women were significantly associated with TB risk. Our results indicate that rs2227982 in *PDCD1* and rs13170556 in *HAVCR2* are associated with increased TB susceptibility in men and that the *CTLA4* haplotype appears protective against TB in women.

## Introduction

Tuberculosis (TB) is an ancient disease primarily caused by *Mycobacterium tuberculosis* (*Mtb*). It remains a socially significant condition and was the thirteenth leading cause of death in 2019 [1]. In recent years, associations of genetic variations in host immune-related genes and TB susceptibility have been reported [2–4], and some research has implicated immune checkpoint proteins in TB risk [5, 6]. These proteins, including programmed cell death-1 (PD-1), cytotoxic T-lymphocyte antigen-4 (CTLA-4), and T-cell immunoglobulin and mucin domain 3 (TIM3), represent the immune regulatory function of co-inhibitory receptors associated

**Funding:** This work was supported by the Taoyuan General Hospital, Taiwan (PTH111024), and the Department of Health and Welfare, Executive Yuan, Taiwan (11107).

**Competing interests:** The authors have declared that no competing interests exist.

with T-cell dysfunction and exhaustion in many cancers [6–8]. Although the possible involvement of immune checkpoint proteins in *Mtb* infection has attracted increasing focus [9–11], the association of their genetic polymorphisms with TB susceptibility is not well understood. Careful examination of the relationship between these polymorphisms and TB risk would increase understanding about human susceptibility to TB.

PD-1 that is encoded by the *PDCD1* gene on human chromosome 2, which consists of 5 exons, is expressed during T-cell activation. Through its ligands–programmed cell death 1 ligand 1 (PD-L1) and PD-L2 it counters positive signals through the T-cell receptor and CD28 [12]. Patients with active TB have higher levels of CD4+ T-cell PD-1 expression compared with unaffected controls [13, 14]. In addition, PD-1$^{-/-}$mice are more susceptible to *Mtb*-induced mortality because of excessive inflammation and uncontrolled bacterial proliferation in the lungs [15]. Also in mice, PD-1 ligation decreases interferon (IFN)-γ production by CD4 + T cells and results in exacerbated pulmonary *Mtb* infection and early host mortality [16]. CD4+ T cells and macrophages enhance phagocytosis and intracellular killing of *Mtb in vitro* though blocking the PD-1/PD-L1 pathway, suggesting an important role for PD-1 in *Mtb* infections [13].

CTLA4 is encoded by the *CTLA4* gene that is located on human chromosome 2 and consists of 4 exons. CTLA-4, a transmembrane homodimer glycoprotein, is expressed by activated effector T cells and negatively regulates T-cell immune responses [17]. Furthermore, patients with latent TB infection with *Mtb* have higher levels of PD-1, CTLA-4, and TIM3 in lymphocytes compared with unaffected controls, possibly because of an evasion of immune exhaustion in the *Mtb* infection [18]. In peripheral blood mononuclear cells from patients with latent TB infection, *Mtb* culture-filtrate antigen was shown to induce production of IFN-γ and interleukin (IL)-17 after CTLA-4 blockade [19].

TIM3, a member of the TIM family of immunoregulatory proteins, is associated with regulation of immune responses in autoimmunity and cancer [20]. TIM3 is encoded by the *HAVCR2* gene on human chromosome 5, which consists of 7 exons. Previous research has indicated increased TIM3 expression in CD8+ T cells in mice during chronic *Mtb* infection [21, 22]. In patients with *Mtb* infection, high levels of TIM3 expression are also found in CD8+ T cells [23, 24]. Work with mouse models has shown that TIM3 influences T-cell exhaustion by decreasing production of IFN-γ and tumor necrosis factor [11]. In chronically infected susceptible mice, blockade of TIM3 restores T-cell function and improves bacterial control [11]. Taken together, these findings suggest an important role for PD-1, CTLA-4, and TIM3 in the host immune response during *Mtb* infection. Despite their involvement, however, the interaction of genetic polymorphisms of *PDCD1*, *CTLA4*, and *HAVCR2* in TB susceptibility is not well known.

Associations of TB susceptibility with some single nucleotide polymorphisms (SNPs) of *PDCD1* and *CTLA4* have been reported [25–27]. In *PDCD1*, rs7568402 is linked to increased TB risk [25], and rs3087243 AA in *CTLA4* is negatively associated with severe pathology in pulmonary TB [26]. In people of Southern Han Chinese ethnicity, rs231775 AG in *CTLA4* has been associated with reduced risk for TB infection [27]. However, whether other SNPs in *PDCD1*, *CTLA4*, and *HAVCR2* are associated with TB risk has not been demonstrated. To address this gap, we investigated the association of 11 SNPs of *PDCD1*, *CTLA4*, and *HAVCR2* genes with TB risk.

## Subjects and methods

### Study population

A total of 285 individuals diagnosed with *Mtb* infection were included in the study conducted at General Taoyuan Hospital in Taoyuan, Taiwan. We gathered medical record information

for outpatients from January 2022 to December 2022. The inclusion criteria were age ≥20 years and diagnosis with active TB disease (i.e., evident TB lesions on X-ray or computed tomography, or positive results of sputum smears and cultures for *Mtb*). As a control group, 270 adult participants without a history of TB disease were enrolled. To avoid the interference of some antiviral and anti-cancer drugs, patients with cancer or other immune-related diseases and viral infections (e.g., hepatitis B, hepatitis C, HIV) were excluded. The study protocol was developed according to the ethical guidelines of the 1975 Declaration of Helsinki and was approved by the Ethics Committee of Taoyuan General Hospital, Taoyuan, Taiwan (TYGH110033). All participants provided written informed consent for this study.

## DNA extraction and SNP selection

Genomic DNA was extracted from peripheral blood cells or oral swabs collected from the enrolled participants respectively using a Quick-DNA™ Miniprep Kit or a QIAamp DNA Mini Kit according to the manufacturer's instructions. Briefly, cell pellets from 300 μl of whole blood or buccal swab were lysed using the respective lysis buffer. After centrifugation at 14,000 ×*g* for 1 min, the supernatant was applied to the respective spin column for DNA purification and washed twice with wash buffer. After sterile distilled deionized water (150 μl) was added in the spin column for a 2-min incubation at room temperature, DNA was eluted by centrifuging at 14,000 ×*g* for 2 min. The extracted genomic DNA was analyzed using agarose gel electrophoresis, quantified by spectrophotometry, and stored at -80˚C until SNP genotyping (Feng Chi Biotech Corp., Taipei, Taiwan). The tSNPs within the genomic regions of *PDCD1*, *CTLA4*, and *HAVCR2*, along with the 1500 base pairs upstream, were chosen using the SeattleSNPs website (http://pga.mbt.washington.edu/education.html) based on data specific to the Han Chinese population in Beijing (HapMap-HCB). The SeattleSNPs database revealed a limited number of polymorphisms (MAF > 0) within our designated region, with four SNPs identified in *PDCD1* and three in *CTLA4*. In the *HAVCR2* region, numerous SNPs were documented and organized into four bins. From these, we opted for four SNPs in *PDCD1*, three in *CTLA4*, and four in *HAVCR2* (one from each bin) for subsequent genotyping. The characteristics of the selected SNPs were presented in S1 Table.

## Genotyping assay

Genotyping of the tag SNPs was performed using the Agena MassARRAY platform with iPLEX reagent chemistry (Agena, San Diego, CA). Briefly, after amplification by PCR of the genomic DNA regions containing the tag SNPs, a single base extension reaction was performed to generate allele-specific diagnostic products. These products were identified by their unique molecular weights, using matrix-assisted laser desorption ionization–time-of-flight mass spectrometry. After allele-specific diagnostic products were loaded onto a matrix pad of a Spectro-CHIP (Agena), the Spectro-CHIP was analyzed using a MassARRAY Analyzer 4, and results were evaluated using clustering analysis with TYPER 4.0 software. The specific primer sequences of PCR and unextended primers used in this study are shown in S2 Table.

## Statistical analysis

The intermarker linkage disequilibrium (LD) of tag SNPs in the three genes was estimated, and haplotype blocks were defined based on $r^2$ and D' values, using Haploview v.4.2. Unpaired Student's *t*-test was used to examine differences in age between non-TB and TB groups. Logistic regression and $\chi^2$ tests were used for association analyses, and odds ratios (ORs) and 95% confidence intervals (CIs) for TB risk were calculated. All statistical assessments were carried out using SPSS 21.0 software (SPSS Inc., Chicago, IL, USA).

## Results

### Participant characteristics

A total of 555 people (430 men and 125 women) were enrolled. As indicated in Table 1, we found no significant difference in sex distribution between the two groups ($p$ = 0.166, $\chi^2$). The mean age of participants with TB was 57 years (range 20–92 years), and the mean age of unaffected controls was 71 years (range 22–99 years). In the overall population and within the sex-defined subgroups, the TB and non-TB groups differed significantly in age ($t$-tests).

### Association of *PDCD1*, *CTLA4*, and *HAVCR2* polymorphisms with TB

When the 11 SNPs in *PDCD1*, *CTLA4*, and *HAVCR2* were genotyped, their genotype frequency distributions were consistent with Hardy–Weinberg equilibrium (Table 2). In the LD plot, one haploblock was identified at *PDCD1* and another at *CTLA4* (Fig 1). Selected SNPs did not differ significantly in genotype frequencies, and TB risk was not associated with these frequencies in the comparison of the non-TB and TB groups in the total population (Table 2).

Logistic regression to evaluate interactions of sex and genotype yielded a significant interaction of sex and the rs13170556 genotype in *HAVCR2* (S3 Table). In addition, genotype and age (<65 vs ≥65 years) showed a significant interaction (S3 Table).

### Association of *PDCD1*, *CTLA4*, and *HAVCR2* polymorphisms with TB in different subgroups

The associations of *PDCD1* rs2227982, *HAVCR2* rs13170556, *CTLA4* rs231775, and *CTLA4* rs231779 with TB susceptibility were sex-dependent. Further sex-stratified analysis showed significant differences in genotype frequencies of rs13170556 between men with and without TB (Table 3). We found that male participants with the TC genotype of *HAVCR2* rs13170556 had a significantly higher risk of TB (adjusted OR [aOR] = 1.824, 95% CI = 1.134–2.935,

**Table 1. Participant characteristics.**

| Variables | TB, N (%) | Non-TB, N (%) | *p* value |
|---|---|---|---|
| **Gender** | | | |
| Man | 214 (75) | 216 (80) | 0.166[a] |
| Woman | 71 (25) | 54 (20) | |
| **Age (years)** | | | |
| Mean ± SD (range) | 57 ± 19 (20–92) | 71 ± 14 (22–99) | <0.001[b] |
| Man, mean ± SD (range) | 59 ± 18 (20–91) | 74 ± 12 (33–99) | <0.001[b] |
| Woman, mean ± SD (range) | 50 ± 20 (20–92) | 60 ± 19 (22–93) | 0.004[b] |
| **Age group-N (%)** | | | |
| < 65 | 184 (65) | 68 (25) | <0.001[a] |
| ≥ 65 | 101 (35) | 202 (75) | |
| **Cavitation-N (%)** | | | |
| Yes | 73 (26) | | |
| No | 212 (74) | | |
| **Pleural effusion-N (%)** | | | |
| Yes | 23 (8) | | |
| No | 262 (92) | | |

Abbreviations: SD, standard deviation; TB, tuberculosis; N, number of participants.

[a]$\chi^2$ tests

[b]$t$-tests

**Table 2. The differences between groups with and without TB in genotypes and alleles frequencies of selected SNPs and results of odds ratio analysis in the overall participants.**

| SNP | Genotype | Counts | | *p* value[a] | *p*$_c$ value | Adj. OR (95% CI)[b] | *p* value for Adj. OR |
| --- | --- | --- | --- | --- | --- | --- | --- |
| | | TB group n = 285 | Non-TB group n = 270 | | | | |
| *PDCD1* | | | | | | | |
| rs10204525 | CC | 25 (9) | 24 (9) | 0.305 | NS | 0.784 (0.396, 1.552) | 0.485 |
| HWp = 0.625 | TC | 133 (47) | 109 (40) | | | 1.262 (0.863, 1.844) | 0.230 |
| | TT (ref.) | 127 (44) | 137 (51) | | | 1 | |
| Allele model | C | 183 (32) | 157 (29) | 0.274 | NS | 1.030 (0.778, 1.362) | 0.839 |
| | T (ref.) | 387 (68) | 383 (71) | | | 1 | |
| Dominant model | TT | 127 (44) | 137 (51) | 0.145 | NS | 0.856 (0.596, 1.229) | 0.399 |
| | TC+CC (ref.) | 158 (56) | 133 (49) | | | 1 | |
| Recessive model | CC | 25 (9) | 24 (9) | 0.961 | NS | 0.703 (0.364, 1.359) | 0.294 |
| | TT+TC (ref.) | 260 (91) | 246 (91) | | | 1 | |
| Overdominant model | TC | 133 (47) | 109 (40) | 0.135 | NS | 1.307 (0.906, 1.885) | 0.152 |
| | TT+CC (ref.) | 152 (53) | 161 (60) | | | 1 | |
| rs2227982 | AA | 61 (21) | 74 (27) | 0.213 | NS | 0.988 (0.594, 1.642) | 0.962 |
| HWp = 0.578 | GA | 147 (52) | 123 (46) | | | 1.430 (0.918, 2.228) | 0.114 |
| | GG (ref.) | 77 (27) | 73 (27) | | | 1 | |
| Allele model | A | 269 (47) | 271 (50) | 0.349 | NS | 0.996 (0.771, 1.287) | 0.978 |
| | G (ref.) | 301 (53) | 269 (50) | | | 1 | |
| Dominant model | GG | 77 (27) | 73 (27) | 0.996 | NS | 0.794 (0.524, 1.203) | 0.276 |
| | AA+GA (ref.) | 208 (73) | 197 (73) | | | 1 | |
| Recessive model | AA | 61 (21) | 74 (27) | 0.099 | NS | 0.783 (0.515, 1.190) | 0.252 |
| | GA+GG (ref.) | 224 (79) | 196 (73) | | | 1 | |
| Overdominant model | GA | 147 (52) | 123 (46) | 0.156 | NS | 1.439 (0.998, 2.074) | 0.051 |
| | AA+GG (ref.) | 138 (48) | 147 (54) | | | 1 | |
| rs7421861 | GG | 6 (2) | 8 (3) | 0.462 | NS | 0.661 (0.202, 2.167) | 0.495 |
| HWp = 0.932 | GA | 85 (30) | 69 (26) | | | 1.132 (0.749, 1.711) | 0.555 |
| | AA (ref.) | 194 (68) | 193 (71) | | | 1 | |
| Allele model | G | 97 (17) | 85 (16) | 0.566 | NS | 1.020 (0.718, 1.451) | 0.910 |
| | A (ref.) | 473 (83) | 455 (84) | | | 1 | |
| Dominant model | AA | 194 (68) | 193 (71) | 0.382 | NS | 0.925 (0.620, 1.379) | 0.702 |
| | GA+GG (ref.) | 91 (32) | 77 (29) | | | 1 | |
| Recessive model | GG | 6 (2) | 8 (3) | 0.520 | NS | 0.639 (0.196, 2.086) | 0.459 |
| | AA+GA (ref.) | 279 (98) | 262 (97) | | | 1 | |
| Overdominant model | GA | 85 (30) | 69 (26) | 0.262 | NS | 1.148 (0.761, 1.732) | 0.511 |
| | AA+GG (ref.) | 200 (70) | 201 (74) | | | 1 | |
| rs6710479 | CC | 20 (7) | 14 (5) | 0.263 | NS | 0.961 (0.430, 2.150) | 0.924 |
| HWp = 0.834 | CT | 116 (41) | 97 (36) | | | 1.262 (0.862, 1.846) | 0.231 |
| | TT (ref.) | 149 (52) | 159 (59) | | | 1 | |
| Allele model | C | 156 (27) | 125 (23) | 0.106 | NS | 1.119 (0.831, 1.508) | 0.458 |
| | T (ref.) | 414 (73) | 415 (77) | | | 1 | |
| Dominant model | TT | 149 (52) | 159 (59) | 0.117 | NS | 0.821 (0.569, 1.183) | 0.289 |
| | CT+CC (ref.) | 136 (48) | 111 (41) | | | 1 | |
| Recessive model | CC | 20 (7) | 14 (5) | 0.368 | NS | 0.876 (0.398, 1.931) | 0.743 |
| | TT+CT (ref.) | 265 (93) | 256 (95) | | | 1 | |
| Overdominant model | CT | 116 (41) | 97 (36) | 0.248 | NS | 1.266 (0.871, 1.840) | 0.216 |
| | TT+CC (ref.) | 169 (59) | 173 (64) | | | 1 | |

*(Continued)*

**Table 2.** (Continued)

| SNP | Genotype | Counts | | *p* value[a] | *p_c* value | Adj. OR (95% CI)[b] | *p* value for Adj. OR |
|---|---|---|---|---|---|---|---|
| | | TB group n = 285 | Non-TB group n = 270 | | | | |
| *CTLA4* | | | | | | | |
| rs231775 | AA | 31 (11) | 29 (11) | 0.565 | NS | 0.698 (0.378, 1.288) | 0.250 |
| HWp = 0.369 | AG | 128 (45) | 133 (49) | | | 0.729 (0.495, 1.073) | 0.109 |
| | GG (ref.) | 126 (44) | 108 (40) | | | 1 | |
| Allele model | A | 190 (33) | 191 (35) | 0.475 | NS | 0.803 (0.613, 1.053) | 0.113 |
| | G (ref.) | 380 (67) | 349 (65) | | | 1 | |
| Dominant model | GG | 126 (44) | 108 (40) | 0.315 | NS | 1.384 (0.957, 2.000) | 0.084 |
| | AA+AG (ref.) | 159 (56) | 162 (60) | | | 1 | |
| Recessive model | AA | 31 (11) | 29 (11) | 0.959 | NS | 0.824 (0.463, 1.467) | 0.510 |
| | AG+GG (ref.) | 254 (89) | 241 (89) | | | 1 | |
| Overdominant model | AG | 128 (45) | 133 (49) | 0.305 | NS | 0.785 (0.546, 1.130) | 0.193 |
| | AA+GG (ref.) | 157 (55) | 137 (51) | | | 1 | |
| rs231777 | TT | 1 (1) | 4 (1) | 0.281 | NS | 0.323 (0.035, 2.987) | 0.319 |
| HWp = 1.000 | TC | 55 (19) | 45 (17) | | | 1.169 (0.729, 1.876) | 0.516 |
| | CC (ref.) | 229 (80) | 221 (82) | | | 1 | |
| Allele model | T | 57 (10) | 53 (10) | 0.918 | NS | 1.028 (0.671, 1.575) | 0.899 |
| | C (ref.) | 513 (90) | 487 (90) | | | 1 | |
| Dominant model | CC | 229 (80) | 221 (82) | 0.652 | NS | 0.907 (0.572, 1.440) | 0.680 |
| | TT+TC (ref.) | 56 (20) | 49 (18) | | | 1 | |
| Recessive model | TT | 1 (1) | 4 (1) | 0.159 | NS | 0.313 (0.034, 2.894) | 0.306 |
| | TC+CC (ref.) | 284 (99) | 266 (99) | | | 1 | |
| Overdominant model | TC | 55 (19) | 45 (17) | 0.420 | NS | 1.184 (0.739, 1.899) | 0.483 |
| | TT+CC (ref.) | 230 (81) | 225 (83) | | | 1 | |
| rs231779 | CC | 31 (11) | 29 (11) | 0.565 | NS | 0.698 (0.378, 1.288) | 0.250 |
| HWp = 0.369 | CT | 128 (45) | 133 (49) | | | 0.729 (0.495, 1.073) | 0.109 |
| | TT (ref.) | 126 (44) | 108 (40) | | | 1 | |
| Allele model | C | 190 (33) | 191 (35) | 0.475 | NS | 0.803 (0.613, 1.053) | 0.113 |
| | T (ref.) | 380 (67) | 349 (65) | | | 1 | |
| Dominant model | TT | 126 (44) | 108 (40) | 0.315 | NS | 1.384 (0.957, 2.000) | 0.084 |
| | CT+CC (ref.) | 159 (56) | 162 (60) | | | 1 | |
| Recessive model | CC | 31 (11) | 29 (11) | 0.959 | NS | 0.824 (0.463, 1.467) | 0.510 |
| | CT+TT (ref.) | 254 (89) | 241 (89) | | | 1 | |
| Overdominant model | CT | 128 (45) | 133 (49) | 0.305 | NS | 0.785 (0.546, 1.130) | 0.193 |
| | TT+CC (ref.) | 157 (55) | 137 (51) | | | 1 | |
| *HAVCR2* | | | | | | | |
| rs9313441 | AA | 0 | 0 | | | | |
| HWp = 0.790 | AG | 22 (8) | 23 (9) | 0.730 | NS | 1.022 (0.524, 1.993) | 0.948 |
| | GG (ref.) | 263 (92) | 247 (91) | | | 1 | |
| Allele model | A | 22 (4) | 23 (4) | 0.736 | NS | 1.021 (0.532, 1.962) | 0.949 |
| | G (ref.) | 548 (96) | 517 (96) | | | 1 | |
| Dominant model | GG | 263 (92) | 247 (91) | 0.730 | NS | 0.978 (0.502, 1.907) | 0.948 |
| | AA+AG (ref.) | 22 (8) | 23 (9) | | | 1 | |
| Recessive model | AA | 0 | 0 | ND | ND | ND | ND |
| | AG+GG (ref.) | 285 (100) | 270 (100) | | | | |
| Overdominant model | AG | 22 (8) | 23 (9) | 0.730 | NS | 1.022 (0.524, 1.993) | 0.948 |

(*Continued*)

**Table 2.** (Continued)

| SNP | Genotype | Counts | | *p* value[a] | *p*c value | Adj. OR (95% CI)[b] | *p* value for Adj. OR |
|---|---|---|---|---|---|---|---|
| | | TB group n = 285 | Non-TB group n = 270 | | | | |
| | AA+GG (ref.) | 263 (92) | 247 (91) | | | 1 | |
| rs13170556 | CC | 13 (5) | 8 (3) | 0.055 | NS | 1.821 (0.671, 4.942) | 0.239 |
| HWp = 0.395 | TC | 91 (32) | 65 (24) | | | 1.375 (0.915, 2.068) | 0.126 |
| | TT (ref.) | 181 (63) | 197 (73) | | | 1 | |
| Allele model | C | 117 (21) | 81 (15) | **0.016** | **0.032** | 1.379 (0.983, 1.934) | 0.063 |
| | T (ref.) | 453 (79) | 459 (85) | | | 1 | |
| Dominant model | TT | 181 (63) | 197 (73) | **0.017** | NS | 0.704 (0.476, 1.042) | 0.079 |
| | TC+CC (ref.) | 104 (37) | 73 (27) | | | 1 | |
| Recessive model | CC | 13 (5) | 8 (3) | 0.324 | NS | 1.664 (0.616, 4.493) | 0.315 |
| | TC+TT (ref.) | 272 (95) | 262 (97) | | | 1 | |
| Overdominant model | TC | 91 (32) | 65 (24) | **0.040** | NS | 1.335 (0.891, 2.002) | 0.161 |
| | TT+CC (ref.) | 194 (68) | 205 (76) | | | 1 | |
| rs919744 | GG | 0 | 0 | | | | |
| HWp = 1.000 | GC | 9 (3) | 7 (3) | 0.691 | NS | 1.135 (0.376, 3.423) | 0.823 |
| | CC (ref.) | 276 (97) | 263 (97) | | | 1 | |
| Allele model | G | 9 (2) | 7 (1) | 0.693 | NS | 1.133 (0.378, 3.390) | 0.824 |
| | C (ref.) | 561 (98) | 533 (99) | | | 1 | |
| Dominant model | CC | 276 (97) | 263 (97) | 0.691 | NS | 0.881 (0.292, 2.659) | 0.823 |
| | GC+GG (ref.) | 9 (3) | 7 (3) | | | 1 | |
| Recessive model | GG | 0 | 0 | ND | ND | ND | ND |
| | GC+CC (ref.) | 285 (100) | 270 (100) | | | | |
| Overdominant model | GC | 9 (3) | 7 (3) | 0.691 | NS | 1.135 (0.376, 3.423) | 0.823 |
| | GG+CC (ref.) | 276 (97) | 263 (97) | | | 1 | |
| rs1036199 | CC | 0 | 0 | | | | |
| HWp = 1.000 | CA | 9 (3) | 7 (3) | 0.691 | NS | 1.135 (0.376, 3.423) | 0.823 |
| | AA (ref.) | 276 (97) | 263 (97) | | | 1 | |
| Allele model | C | 9 (2) | 7 (1) | 0.693 | NS | 1.133 (0.378, 3.390) | 0.824 |
| | A (ref.) | 561 (98) | 533 (99) | | | 1 | |
| Dominant model | AA | 276 (97) | 263 (97) | 0.691 | NS | 0.881 (0.292, 2.659) | 0.823 |
| | CA+CC (ref.) | 9 (3) | 7 (3) | | | 1 | |
| Recessive model | CC | 0 | 0 | ND | ND | ND | ND |
| | CA+AA (ref.) | 285 (100) | 270 (100) | | | | |
| Overdominant model | CA | 9 (3) | 7 (3) | 0.691 | NS | 1.135 (0.376, 3.423) | 0.823 |
| | AA+CC (ref.) | 276 (97) | 263 (97) | | | 1 | |

Abbreviations: Ref., reference genotype; CI, confidence interval; OR, odds ratio; Pc, the Bonferroni correction of P values; HWp, p value of Hardy-Weinberg disequilibrium test.

[a] $\chi^2$ test.

[b] Adj. = adjusted for age and sex by logistic regression.

*p* = 0.013) compared with those who had the TT genotype (Table 3). In addition, *HAVCR2* rs13170556 and TB risk were significantly associated under the allele model (aOR = 1.606, 95% CI = 1.077–2.394, *p* = 0.020), the dominant model (aOR = 0.556, 95% CI = 0.351–0.879, *p* = 0.012), and the overdominant model (aOR = 1.790, 95% CI = 1.116–2.871, *p* = 0.016). The overdominant model also showed a significant association between *PDCD1* rs2227982 and TB risk (aOR = 1.642, 95% CI = 1.070–2.519, *p* = 0.023) (Table 3).

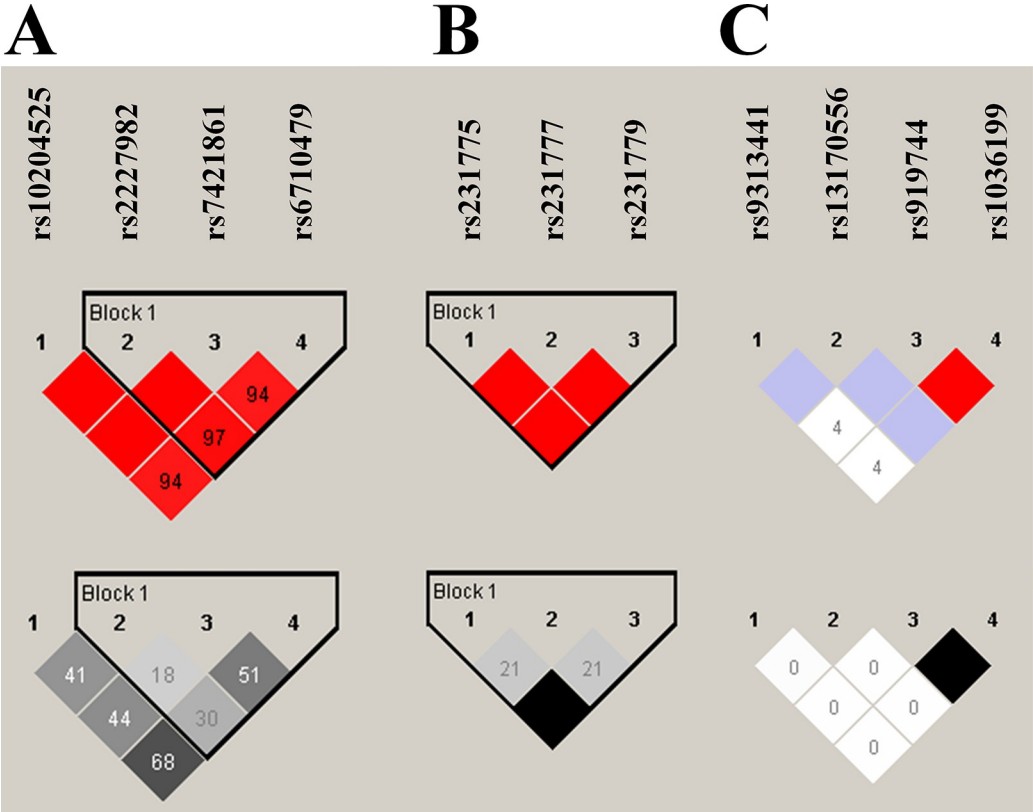

**Fig 1. Linkage disequilibrium plot in D' demonstrating adjacent strength between SNP pairs in *PDCD1* (10204525, rs2227982, rs7421861, and rs6710479), *CTLA4* (rs231775, rs231777, and rs231779), and *HAVCR2* (rs9313441, rs13170556, rs919744, and rs1036199).** One haploblock was identified in *PDCD1* (A) and another in *CTLA4* (B).

Female participants with the heterozygous *CTLA4* AG genotype of rs231775 or heterozygous CT genotype of rs231779 had significantly lower risk of TB (aOR = 0.421, 95% CI = 0.188–0.944, *p* = 0.036) compared with female participants who had the GG or TT genotype (Table 4). In addition, under the dominant model, significant associations of rs231775 and rs231779 with TB risk were observed in female population (aOR = 2.428, 95% CI = 1.107–5.323, *p* = 0.027) (Table 4).

We further evaluated the association of TB risk with the combination of *PDCD1* rs2227982 and *HAVCR2* rs13170556 in the male population. Male participants with the GA/TC combination had a 2.9-fold increased risk of TB (aOR = 2.926, 95% CI = 1.335–6.411, *p* = 0.007). In addition, double heterozygosity (*PDCD1*/*HAVCR2* GA/CT vs all remaining allele combinations) was still associated with a 3.1-fold increased risk of TB (aOR = 3.064, 95% CI = 1.593–5.891, *p* = 0.001) (Table 5).

In age-stratified groups, genotype frequencies of the 11 SNPs did not differ significantly between those <65 years and those ≥65 years (S4 and S5 Tables). In addition, in these two groups, SNPs were not significantly associated with TB risk (S4 and S5 Tables).

## Association of haplotypes of *PDCD1* and *CTLA4* with TB risk

The results of LD analysis for polymorphisms of *PDCD1*, *CTLA4*, and *HAVCR2* are presented in Fig 1. Of the eight possible haplotypes, six haplotypes of *PDCD1* rs2227982-rs7421861-rs6710479 and four haplotypes of *CTLA4* rs231775-rs231777-rs231779

**Table 3. The differences between groups with and without TB in genotypes and alleles frequencies of selected SNPs and results of odds ratio analysis in male participants.**

| SNP | Genotype | Counts | | p value[a] | p_c value | Adj. OR (95% CI)[b] | p value for Adj. OR |
|---|---|---|---|---|---|---|---|
| | | TB group n = 214 | Non-TB group n = 216 | | | | |
| *PDCD1* | | | | | | | |
| rs10204525 | CC | 17 (8) | 19 (9) | 0.351 | NS | 0.826 (0.371, 1.837) | 0.639 |
| | TC | 101 (47) | 87 (40) | | | 1.358 (0.873, 2.112) | 0.175 |
| | TT (ref.) | 96 (45) | 110 (51) | | | 1 | |
| Allele model | C | 135 (32) | 125 (29) | 0.405 | NS | 1.079 (0.778, 1.495) | 0.649 |
| | T (ref.) | 293 (68) | 307 (71) | | | 1 | |
| Dominant model | TT | 96 (45) | 110 (51) | 0.208 | NS | 0.799 (0.524, 1.217) | 0.295 |
| | TC+CC (ref.) | 118 (55) | 106 (49) | | | 1 | |
| Recessive model | CC | 17 (8) | 19 (9) | 0.750 | NS | 0.718 (0.331, 1.555) | 0.401 |
| | TT+TC (ref.) | 197 (92) | 197 (91) | | | 1 | |
| Overdominant model | TC | 101 (47) | 87 (40) | 0.148 | NS | 1.395 (0.910, 2.139) | 0.127 |
| | TT+CC (ref.) | 113 (53) | 129 (60) | | | 1 | |
| rs2227982 | AA | 46 (22) | 61 (28) | 0.180 | NS | 0.887 (0.494, 1.591) | 0.687 |
| | GA | 110 (51) | 94 (44) | | | 1.548 (0.925, 2.589) | 0.096 |
| | GG (ref.) | 58 (27) | 61 (28) | | | 1 | |
| Allele model | A | 202 (47) | 216 (50) | 0.411 | NS | 0.943 (0.701, 1.270) | 0.701 |
| | G (ref.) | 226 (53) | 216 (50) | | | 1 | |
| Dominant model | GG | 58 (27) | 61 (28) | 0.792 | NS | 0.789 (0.490, 1.271) | 0.330 |
| | AA+GA (ref.) | 156 (73) | 155 (72) | | | 1 | |
| Recessive model | AA | 46 (22) | 61 (28) | 0.106 | NS | 0.673 (0.415, 1.093) | 0.110 |
| | GA+GG (ref.) | 168 (78) | 155 (72) | | | 1 | |
| Overdominant model | GA | 110 (51) | 94 (44) | 0.102 | NS | 1.642 (1.070, 2.519) | **0.023** |
| | AA+GG (ref.) | 104 (49) | 122 (56) | | | 1 | |
| rs7421861 | GG | 4 (2) | 8 (4) | 0.264 | NS | 0.555 (0.142, 2.170) | 0.397 |
| | GA | 62 (29) | 51 (23) | | | 1.397 (0.856, 2.279) | 0.181 |
| | AA (ref.) | 148 (69) | 157 (73) | | | 1 | |
| Allele model | G | 70 (16) | 67 (16) | 0.735 | NS | 1.128 (0.747, 1.703) | 0.567 |
| | A (ref.) | 358 (84) | 365 (84) | | | 1 | |
| Dominant model | AA | 148 (69) | 157 (73) | 0.421 | NS | 0.784 (0.490, 1.254) | 0.310 |
| | GA+GG (ref.) | 66 (31) | 59 (27) | | | 1 | |
| Recessive model | GG | 4 (2) | 8 (4) | 0.248 | NS | 0.509 (0.131, 1.975) | 0.329 |
| | AA+GA (ref.) | 210 (98) | 208 (96) | | | 1 | |
| Overdominant model | GA | 62 (29) | 51 (23) | 0.207 | NS | 1.427 (0.876, 2.323) | 0.153 |
| | AA+GG (ref.) | 152 (71) | 165 (77) | | | 1 | |
| rs6710479 | CC | 12 (5) | 13 (6) | 0.684 | NS | 0.617 (0.235, 1.623) | 0.328 |
| | CT | 85 (40) | 77 (36) | | | 1.317 (0.843, 2.056) | 0.226 |
| | TT (ref.) | 117 (55) | 126 (58) | | | 1 | |
| Allele model | C | 109 (25) | 103 (24) | 0.580 | NS | 1.041 (0.734, 1.476) | 0.822 |
| | T (ref.) | 319 (75) | 329 (76) | | | 1 | |
| Dominant model | TT | 117 (55) | 126 (58) | 0.444 | NS | 0.836 (0.546, 1.281) | 0.411 |
| | CT+CC (ref.) | 97 (45) | 90 (42) | | | 1 | |
| Recessive model | CC | 12 (5) | 13 (6) | 0.855 | NS | 0.557 (0.215, 1.441) | 0.227 |
| | TT+CT (ref.) | 202 (95) | 203 (94) | | | 1 | |
| Overdominant model | CT | 85 (40) | 77 (36) | 0.384 | NS | 1.369 (0.883, 2.121) | 0.161 |
| | TT+CC (ref.) | 129 (60) | 139 (64) | | | 1 | |

*(Continued)*

**Table 3.** (Continued)

| SNP | Genotype | Counts | | *p* value[a] | *p*<sub>c</sub> value | Adj. OR (95% CI)[b] | *p* value for Adj. OR |
|---|---|---|---|---|---|---|---|
| | | TB group n = 214 | Non-TB group n = 216 | | | | |
| *CTLA4* | | | | | | | |
| rs231775 | AA | 27 (13) | 24 (11) | 0.838 | NS | 0.780 (0.392, 1.550) | 0.478 |
| | AG | 96 (45) | 102 (47) | | | 0.897 (0.572, 1.405) | 0.634 |
| | GG (ref.) | 91 (42) | 90 (42) | | | 1 | |
| Allele model | A | 150 (35) | 150 (35) | 0.920 | NS | 0.886 (0.647, 1.213) | 0.450 |
| | G (ref.) | 278 (65) | 282 (65) | | | 1 | |
| Dominant model | GG | 91 (42) | 90 (42) | 0.857 | NS | 1.149 (0.751, 1.757) | 0.522 |
| | AA+AG (ref.) | 123 (58) | 126 (58) | | | 1 | |
| Recessive model | AA | 27 (13) | 24 (11) | 0.629 | NS | 0.824 (0.431, 1.575) | 0.558 |
| | AG+GG (ref.) | 187 (87) | 192 (89) | | | 1 | |
| Overdominant model | AG | 96 (45) | 102 (47) | 0.623 | NS | 0.946 (0.620, 1.445) | 0.798 |
| | AA+GG (ref.) | 118 (55) | 114 (53) | | | 1 | |
| rs231777 | TT | 1 (1) | 4 (2) | 0.315 | NS | 0.340 (0.036, 3.196) | 0.345 |
| | TC | 43 (20) | 37 (17) | | | 1.230 (0.718, 2.104) | 0.451 |
| | CC (ref.) | 170 (79) | 175 (81) | | | 1 | |
| Allele model | T | 45 (11) | 45 (10) | 0.963 | NS | 1.047 (0.649, 1.688) | 0.852 |
| | C (ref.) | 383 (89) | 387 (90) | | | 1 | |
| Dominant model | CC | 170 (79) | 175 (81) | 0.681 | NS | 0.876 (0.519, 1.477) | 0.619 |
| | TT+TC (ref.) | 44 (21) | 41 (19) | | | 1 | |
| Recessive model | TT | 1 (1) | 4 (2) | 0.181 | NS | 0.327 (0.035, 3.063) | 0.327 |
| | TC+CC (ref.) | 213 (99) | 212 (98) | | | 1 | |
| Overdominant model | TC | 43 (20) | 37 (17) | 0.430 | NS | 1.249 (0.730, 2.136) | 0.418 |
| | TT+CC (ref.) | 171 (80) | 179 (83) | | | 1 | |
| rs231779 | CC | 27 (13) | 24 (11) | 0.838 | NS | 0.780 (0.392, 1.550) | 0.478 |
| | CT | 96 (45) | 102 (47) | | | 0.897 (0.572, 1.405) | 0.634 |
| | TT (ref.) | 91 (42) | 90 (42) | | | 1 | |
| Allele model | C | 150 (35) | 150 (35) | 0.920 | NS | 0.886 (0.647, 1.213) | 0.450 |
| | T (ref.) | 278 (65) | 282 (65) | | | 1 | |
| Dominant model | TT | 91 (42) | 90 (42) | 0.857 | NS | 1.149 (0.751, 1.757) | 0.522 |
| | CT+CC (ref.) | 123 (58) | 126 (58) | | | 1 | |
| Recessive model | CC | 27 (13) | 24 (11) | 0.629 | NS | 0.824 (0.431, 1.575) | 0.558 |
| | CT+TT (ref.) | 187 (87) | 192 (89) | | | 1 | |
| Overdominant model | CT | 96 (45) | 102 (47) | 0.623 | NS | 0.946 (0.620, 1.445) | 0.798 |
| | TT+CC (ref.) | 118 (55) | 114 (53) | | | 1 | |
| *HAVCR2* | | | | | | | |
| rs9313441 | AA | 0 | 0 | | | ND | ND |
| | AG | 13 (6) | 18 (8) | 0.365 | NS | 0.655 (0.287, 1.494) | 0.315 |
| | GG (ref.) | 201 (94) | 198 (92) | | | 1 | |
| Allele model | A | 13 (3) | 18 (4) | 0.374 | NS | 0.666 (0.297, 1.494) | 0.324 |
| | G (ref.) | 415 (97) | 414 (96) | | | 1 | |
| Dominant model | GG | 201 (94) | 198 (92) | 0.365 | NS | 1.526 (0.669, 3.479) | 0.315 |
| | AA+AG (ref.) | 13 (6) | 18 (8) | | | 1 | |
| Recessive model | AA | 0 | 0 | ND | ND | ND | ND |
| | AG+GG (ref.) | 214 (100) | 216 (100) | | | | |
| Overdominant model | AG | 13 (6) | 18 (8) | 0.365 | NS | 0.655 (0.287, 1.494) | 0.315 |

*(Continued)*

**Table 3.** (Continued)

| SNP | Genotype | Counts | | p value[a] | p_c value | Adj. OR (95% CI)[b] | p value for Adj. OR |
| --- | --- | --- | --- | --- | --- | --- | --- |
| | | TB group n = 214 | Non-TB group n = 216 | | | | |
| | AA+GG (ref.) | 201 (94) | 198 (92) | | | 1 | |
| rs13170556 | CC | 7 (3) | 6 (3) | **0.010** | **0.030** | 1.584 (0.452, 5.553) | 0.472 |
| | TC | 74 (35) | 47 (22) | | | 1.824 (1.134, 2.935) | **0.013** |
| | TT (ref.) | 133 (62) | 163 (75) | | | 1 | |
| Allele model | C | 88 (21) | 59 (14) | **0.007** | **0.014** | 1.606 (1.077, 2.394) | **0.020** |
| | T (ref.) | 340 (79) | 373 (86) | | | 1 | |
| Dominant model | TT | 133 (62) | 163 (75) | **0.003** | **0.009** | 0.556 (0.351, 0.879) | **0.012** |
| | TC+CC (ref.) | 81 (38) | 53 (25) | | | 1 | |
| Recessive model | CC | 7 (3) | 6 (3) | 0.765 | NS | 1.332 (0.381, 4.650) | 0.653 |
| | TC+TT (ref.) | 207 (97) | 210 (97) | | | 1 | |
| Overdominant model | TC | 74 (35) | 47 (22) | **0.003** | **0.009** | 1.790 (1.116, 2.871) | **0.016** |
| | TT+CC (ref.) | 140 (65) | 169 (78) | | | 1 | |
| rs919744 | GG | 0 | 0 | | | ND | ND |
| | GC | 8 (4) | 4 (2) | 0.235 | NS | 1.430 (0.358, 5.709) | 0.613 |
| | CC (ref.) | 206 (96) | 212 (98) | | | 1 | |
| Allele model | G | 8 (2) | 4 (1) | 0.238 | NS | 1.423 (0.359, 5.633) | 0.615 |
| | C (ref.) | 420 (98) | 428 (99) | | | 1 | |
| Dominant model | CC | 206 (96) | 212 (98) | 0.235 | NS | 0.699 (0.175, 2.792) | 0.613 |
| | GC+GG (ref.) | 8 (4) | 4 (2) | | | 1 | |
| Recessive model | GG | 0 | 0 | ND | ND | ND | ND |
| | GC+CC (ref.) | 214 (100) | 216 (100) | | | | |
| Overdominant model | GC | 8 (4) | 4 (2) | 0.235 | NS | 1.430 (0.358, 5.709) | 0.613 |
| | GG+CC (ref.) | 206 (96) | 212 (98) | | | 1 | |
| rs1036199 | CC | 0 | 0 | | | ND | ND |
| | CA | 8 (4) | 4 (2) | 0.235 | NS | 1.430 (0.358, 5.709) | 0.613 |
| | AA (ref.) | 206 (96) | 212 (98) | | | 1 | |
| Allele model | C | 8 (2) | 4 (1) | 0.238 | NS | 1.423 (0.359, 5.633) | 0.615 |
| | A (ref.) | 420 (98) | 428 (99) | | | 1 | |
| Dominant model | AA | 206 (96) | 212 (98) | 0.235 | NS | 0.699 (0.175, 2.792) | 0.613 |
| | CA+CC (ref.) | 8 (4) | 4 (2) | | | 1 | |
| Recessive model | CC | 0 | 0 | ND | ND | ND | ND |
| | CA+AA (ref.) | 214 (100) | 216 (100) | | | | |
| Overdominant model | CA | 8 (4) | 4 (2) | 0.235 | NS | 1.430 (0.358, 5.709) | 0.613 |
| | AA+CC (ref.) | 206 (96) | 212 (98) | | | 1 | |

Abbreviations: Ref., reference genotype; CI, confidence interval; OR, odds ratio; Pc, the Bonferroni correction of P values.

[a]$\chi^2$ test.

[b]Adj. = adjusted for age by logistic regression.

were detected (Table 6). In comparison with the most common haplotype of *PDCD1* (A-A-T haplotype of rs2227982-rs7421861-rs6710479) or *CTLA4* (G-C-T haplotype of rs231775-rs231777-rs231779), no haplotype was significantly associated with TB risk overall, among male participants, or in the age-stratified populations (Table 6 and S6 Table). Female participants with the A-C-C haplotype of *CTLA4* had significantly lower risk of TB (aOR = 0.488, 95% CI = 0.263–0.905, $p$ = 0.023) compared with those who had the most common haplotype (G-C-T) of rs231775-rs231777-rs231779 (Table 6).

**Table 4. The differences between groups with and without TB in genotypes and alleles frequencies of selected SNPs and results of odds ratio analysis in female participants.**

| SNP | Genotype | Counts | | *p* value[a] | *p*$_c$ value | Adj. OR (95% CI)[b] | *p* value for Adj. OR |
|---|---|---|---|---|---|---|---|
| | | TB group n = 71 | Non-TB group n = 54 | | | | |
| *PDCD1* | | | | | | | |
| rs10204525 | CC | 8 (11) | 5 (9) | 0.772 | NS | 0.884 (0.236, 3.307) | 0.854 |
| | TC | 32 (45) | 22 (41) | | | 1.123 (0.516, 2.441) | 0.770 |
| | TT (ref.) | 31 (44) | 27 (50) | | | 1 | |
| Allele model | C | 48 (34) | 32 (30) | 0.483 | NS | 1.007 (0.572, 1.773) | 0.980 |
| | T (ref.) | 94 (66) | 76 (70) | | | 1 | |
| Dominant model | TT | 31 (44) | 27 (50) | 0.482 | NS | 0.927 (0.441, 1.949) | 0.842 |
| | TC+CC (ref.) | 40 (56) | 27 (50) | | | 1 | |
| Recessive model | CC | 8 (11) | 5 (9) | 0.716 | NS | 0.833 (0.236, 2.932) | 0.775 |
| | TT+TC (ref.) | 63 (89) | 49 (91) | | | 1 | |
| Overdominant model | TC | 32 (45) | 22 (41) | 0.628 | NS | 1.147 (0.547, 2.406) | 0.716 |
| | TT+CC (ref.) | 39 (55) | 32 (59) | | | 1 | |
| rs2227982 | AA | 15 (21) | 13 (24) | 0.824 | NS | 1.109 (0.366, 3.362) | 0.885 |
| | GA | 37 (52) | 29 (54) | | | 1.058 (0.421, 2.656) | 0.905 |
| | GG (ref.) | 19 (27) | 12 (22) | | | 1 | |
| Allele model | A | 67 (47) | 55 (51) | 0.558 | NS | 1.048 (0.619, 1.774) | 0.862 |
| | G (ref.) | 75 (53) | 53 (49) | | | 1 | |
| Dominant model | GG | 19 (27) | 12 (22) | 0.561 | NS | 0.933 (0.385, 2.262) | 0.878 |
| | AA+GA (ref.) | 52 (73) | 42 (78) | | | 1 | |
| Recessive model | AA | 15 (21) | 13 (24) | 0.695 | NS | 1.065 (0.440, 2.579) | 0.889 |
| | GA+GG (ref.) | 56 (79) | 41 (76) | | | 1 | |
| Overdominant model | GA | 37 (52) | 29 (54) | 0.860 | NS | 1.004 (0.482, 2.094) | 0.991 |
| | AA+GG (ref.) | 34 (48) | 25 (46) | | | 1 | |
| rs7421861 | GG | 2 (3) | 0 | 0.462 | NS | ND | ND |
| | GA | 23 (32) | 18 (33) | | | 0.797 (0.357, 1.776) | 0.578 |
| | AA (ref.) | 46 (65) | 36 (67) | | | 1 | |
| Allele model | G | 27 (19) | 18 (17) | 0.632 | NS | 0.946 (0.474, 1.889) | 0.875 |
| | A (ref.) | 115 (81) | 90 (83) | | | 1 | |
| Dominant model | AA | 46 (65) | 36 (67) | 0.827 | NS | 1.185 (0.534, 2.626) | 0.677 |
| | GA+GG (ref.) | 25 (35) | 18 (33) | | | 1 | |
| Recessive model | GG | 2 (3) | 0 | 0.214 | NS | ND | ND |
| | AA+GA (ref.) | 69 (97) | 54 (100) | | | | |
| Overdominant model | GA | 23 (32) | 18 (33) | 0.912 | NS | 0.765 (0.343, 1.703) | 0.512 |
| | AA+GG (ref.) | 48 (68) | 36 (67) | | | 1 | |
| rs6710479 | CC | 8 (11) | 1 (2) | 0.060 | NS | 5.608 (0.637, 49.349) | 0.120 |
| | CT | 31 (44) | 20 (37) | | | 1.395 (0.647, 3.006) | 0.395 |
| | TT (ref.) | 32 (45) | 33 (61) | | | 1 | |
| Allele model | C | 47 (33) | 22 (20) | **0.026** | NS | 1.629 (0.889, 2.986) | 0.114 |
| | T (ref.) | 95 (67) | 86 (80) | | | 1 | |
| Dominant model | TT | 32 (45) | 33 (61) | 0.075 | NS | 0.626 (0.296, 1.324) | 0.221 |
| | CT+CC (ref.) | 39 (55) | 21 (39) | | | 1 | |
| Recessive model | CC | 8 (11) | 1 (2) | 0.044 | NS | 4.791 (0.560, 40.988) | 0.153 |
| | TT+CT (ref.) | 63 (89) | 53 (98) | | | 1 | |
| Overdominant model | CT | 31 (44) | 20 (37) | 0.455 | NS | 1.190 (0.561, 2.521) | 0.651 |
| | TT+CC (ref.) | 40 (56) | 34 (63) | | | 1 | |

*(Continued)*

**Table 4.** (Continued)

| SNP | Genotype | Counts | | *p* value[a] | *p*<sub>c</sub> value | Adj. OR (95% CI)[b] | *p* value for Adj. OR |
|---|---|---|---|---|---|---|---|
| | | TB group n = 71 | Non-TB group n = 54 | | | | |
| *CTLA4* | | | | | | | |
| rs231775 | AA | 4 (6) | 5 (9) | 0.189 | NS | 0.355 (0.080, 1.569) | 0.172 |
| | AG | 32 (45) | 31 (58) | | | 0.421 (0.188, 0.944) | **0.036** |
| | GG (ref.) | 35 (49) | 18 (33) | | | 1 | |
| Allele model | A | 40 (28) | 41 (38) | 0.101 | NS | 0.573 (0.328, 1.000) | 0.050 |
| | G (ref.) | 102 (72) | 67 (62) | | | 1 | |
| Dominant model | GG | 35 (49) | 18 (33) | 0.074 | NS | 2.428 (1.107, 5.323) | **0.027** |
| | AA+AG (ref.) | 36 (51) | 36 (67) | | | 1 | |
| Recessive model | AA | 4 (6) | 5 (9) | 0.437 | NS | 0.578 (0.143, 2.340) | 0.442 |
| | AG+GG (ref.) | 67 (94) | 49 (91) | | | 1 | |
| Overdominant model | AG | 32 (45) | 31 (58) | 0.172 | NS | 0.499 (0.234, 1.067) | 0.073 |
| | AA+GG (ref.) | 39 (55) | 23 (42) | | | 1 | |
| rs231777 | TT | 0 | 0 | | | ND | ND |
| | TC | 12 (17) | 8 (15) | 0.753 | NS | 1.048 (0.380, 2.893) | 0.928 |
| | CC (ref.) | 59 (83) | 46 (85) | | | 1 | |
| Allele model | T | 12 (8) | 8 (7) | 0.763 | NS | 1.044 (0.395, 2.756) | 0.931 |
| | C (ref.) | 130 (92) | 100 (93) | | | 1 | |
| Dominant model | CC | 59 (83) | 46 (85) | 0.753 | NS | 0.954 (0.346, 2.635) | 0.928 |
| | TT+TC (ref.) | 12 (17) | 8 (15) | | | 1 | |
| Recessive model | TT | 0 | 0 | ND | ND | ND | ND |
| | TC+CC (ref.) | 71 (100) | 54 (100) | | | | |
| Overdominant model | TC | 12 (17) | 8 (15) | 0.753 | NS | 1.048 (0.380, 2.893) | 0.928 |
| | TT+CC (ref.) | 59 (83) | 46 (85) | | | 1 | |
| rs231779 | CC | 4 (6) | 5 (9) | 0.189 | NS | 0.355 (0.080, 1.569) | 0.172 |
| | CT | 32 (45) | 31 (58) | | | 0.421 (0.188, 0.944) | **0.036** |
| | TT (ref.) | 35 (49) | 18 (33) | | | 1 | |
| Allele model | C | 40 (28) | 41 (38) | 0.101 | NS | 0.573 (0.328, 1.000) | 0.050 |
| | T (ref.) | 102 (72) | 67 (62) | | | 1 | |
| Dominant model | TT | 35 (49) | 18 (33) | 0.074 | NS | 2.428 (1.107, 5.323) | **0.027** |
| | CT+CC (ref.) | 36 (51) | 36 (67) | | | 1 | |
| Recessive model | CC | 4 (6) | 5 (9) | 0.437 | NS | 0.578 (0.143, 2.340) | 0.442 |
| | CT+TT (ref.) | 67 (94) | 49 (91) | | | 1 | |
| Overdominant model | CT | 32 (45) | 31 (58) | 0.172 | NS | 0.499 (0.234, 1.067) | 0.073 |
| | TT+CC (ref.) | 39 (55) | 23 (42) | | | 1 | |
| *HAVCR2* | | | | | | | |
| rs9313441 | AA | 0 | 0 | | | ND | ND |
| | AG | 9 (13) | 5 (9) | 0.548 | NS | 2.060 (0.601, 7.057) | 0.250 |
| | GG (ref.) | 62 (87) | 49 (91) | | | 1 | |
| Allele model | A | 9 (6) | 5 (5) | 0.561 | NS | 1.942 (0.599, 6.294) | 0.269 |
| | G (ref.) | 133 (94) | 103 (95) | | | 1 | |
| Dominant model | GG | 62 (87) | 49 (91) | 0.548 | NS | 0.485 (0.142, 1.663) | 0.250 |
| | AA+AG (ref.) | 9 (13) | 5 (9) | | | 1 | |
| Recessive model | AA | 0 | 0 | ND | ND | ND | ND |
| | AG+GG (ref.) | 71 (100) | 54 (100) | | | | |
| Overdominant model | AG | 9 (13) | 5 (9) | 0.548 | NS | 2.060 (0.601, 7.057) | 0.250 |

(*Continued*)

**Table 4.** (Continued)

| SNP | Genotype | Counts | | *p* value[a] | *p*$_c$ value | Adj. OR (95% CI)[b] | *p* value for Adj. OR |
|---|---|---|---|---|---|---|---|
| | | TB group<br>n = 71 | Non-TB group<br>n = 54 | | | | |
| | AA+GG (ref.) | 62 (87) | 49 (91) | | | 1 | |
| rs13170556 | CC | 6 (8) | 2 (4) | 0.342 | NS | 2.104 (0.375, 11.817) | 0.398 |
| | TC | 17 (24) | 18 (33) | | | 0.559 (0.241, 1.295) | 0.175 |
| | TT (ref.) | 48 (68) | 34 (63) | | | 1 | |
| Allele model | C | 29 (20) | 22 (20) | 0.992 | NS | 0.914 (0.479, 1.744) | 0.784 |
| | T (ref.) | 113 (80) | 86 (80) | | | 1 | |
| Dominant model | TT | 48 (68) | 34 (63) | 0.588 | NS | 1.420 (0.651, 3.097) | 0.378 |
| | TC+CC (ref.) | 23 (32) | 20 (37) | | | 1 | |
| Recessive model | CC | 6 (8) | 2 (4) | 0.283 | NS | 2.485 (0.454, 13.610) | 0.294 |
| | TC+TT (ref.) | 65 (92) | 52 (96) | | | 1 | |
| Overdominant model | TC | 17 (24) | 18 (33) | 0.247 | NS | 0.527 (0.230, 1.210) | 0.131 |
| | TT+CC (ref.) | 54 (76) | 36 (67) | | | 1 | |
| rs919744 | GG | 0 | 0 | | | ND | ND |
| | GC | 1 (1) | 3 (6) | 0.192 | NS | 0.361 (0.035, 3.699) | 0.391 |
| | CC (ref.) | 70 (99) | 51 (94) | | | 1 | |
| Allele model | G | 1 (1) | 3 (3) | 0.196 | NS | 0.370 (0.037, 3.707) | 0.398 |
| | C (ref.) | 141 (99) | 105 (97) | | | 1 | |
| Dominant model | CC | 70 (99) | 51 (94) | 0.192 | NS | 2.771 (0.270, 28.400) | 0.391 |
| | GC+GG (ref.) | 1 (1) | 3 (6) | | | 1 | |
| Recessive model | GG | 0 | 0 | ND | ND | ND | ND |
| | GC+CC (ref.) | 71 (100) | 54 (100) | | | | |
| Overdominant model | GC | 1 (1) | 3 (6) | 0.192 | NS | 0.361 (0.035, 3.699) | 0.391 |
| | GG+CC (ref.) | 70 (99) | 51 (94) | | | 1 | |
| rs1036199 | CC | 0 | 0 | | | ND | ND |
| | CA | 1 (1) | 3 (6) | 0.192 | NS | 0.361 (0.035, 3.699) | 0.391 |
| | AA (ref.) | 70 (99) | 51 (94) | | | 1 | |
| Allele model | C | 1 (1) | 3 (3) | 0.196 | NS | 0.370 (0.037, 3.707) | 0.398 |
| | A (ref.) | 141 (99) | 105 (97) | | | 1 | |
| Dominant model | AA | 70 (99) | 51 (94) | 0.192 | NS | 2.771 (0.270, 28.400) | 0.391 |
| | CA+CC (ref.) | 1 (1) | 3 (6) | | | 1 | |
| Recessive model | CC | 0 | 0 | ND | ND | ND | ND |
| | CA+AA (ref.) | 71 (100) | 54 (100) | | | | |
| Overdominant model | CA | 1 (1) | 3 (6) | 0.192 | NS | 0.361 (0.035, 3.699) | 0.391 |
| | AA+CC (ref.) | 70 (99) | 51 (94) | | | 1 | |

Abbreviations: Ref., reference genotype; CI, confidence interval; OR, odds ratio; Pc, the Bonferroni correction of P values.

[a]$\chi^2$ test.

[b]Adj. = adjusted for age by logistic regression.

## Discussion

Previous groups have reported that some polymorphisms of *PDCD1*, *CTLA4*, and *HAVCR2* are associated with TB risk [25–28]. In this study, we found a sex-dependent association of TB risk with the AG genotype of *PDCD1* rs2227982, TC genotype of *HAVCR2* rs13170556, AG genotype of *CTLA4* rs231775, and CT genotype of *CTLA4* rs231779. In men, the AG/TC combination of *PDCD1*/*HAVCR2* was significantly associated with TB risk. In women, our

**Table 5. Combined association of TB risk and *PDCD1* rs2227982 and *HAVCR2* rs13170556 polymorphisms in male participants.**

| Genotype combinations | TB group n = 214 | Non-TB group n = 216 | *p* value[a] | *p$_c$* value | Adj. OR (95% CI)[a] | *p* value for Adj. OR |
|---|---|---|---|---|---|---|
| rs2227982/rs13170556 | | | | | | |
| GG/TT (Ref.) | 40 (19) | 43 (20) | **0.044** | NS | 1 | |
| GG/TC | 16 (8) | 16 (7) | | | 0.727 (0.289, 1.834) | 0.500 |
| GG/CC | 2 (1) | 2 (1) | | | 1.274 (0.139, 11.684) | 0.830 |
| AG/TT | 67 (31) | 73 (34) | | | 1.088 (0.588, 2.014) | 0.788 |
| AG/TC | 40 (19) | 18 (8) | | | 2.926 (1.335, 6.411) | **0.007** |
| AG/CC | 3 (1) | 3 (1) | | | 1.129 (0.161, 7.927) | 0.903 |
| AA/TT | 26 (12) | 47 (22) | | | 0.608 (0.297, 1.244) | 0.173 |
| AA/TC | 18 (8) | 13 (6) | | | 1.427 (0.571, 3.565) | 0.447 |
| AA/CC | 2 (1) | 1 (1) | | | 2.944 (0.188, 46.006) | 0.441 |
| other combinations (Ref.) | 174 (81) | 198 (92) | **0.002** | **0.018** | 1 | |
| AG/TC | 40 (19) | 18 (8) | | | 3.064 (1.593, 5.891) | **0.001** |

Abbreviations: Ref., reference genotype combination; CI, confidence interval; OR, odds ratio; Pc, the Bonferroni correction of P values.

[a]$\chi^2$-test

[b]Adjusted for age by logistic regression.

haplotype analysis showed that the A-C-C haplotype of *CTLA4* rs231775-rs231777-rs231779 was associated with reduced susceptibility to TB. Our findings imply that genetic polymorphisms of *PDCD1*, *CTLA4*, and *HAVCR2* have an important role in TB susceptibility.

In men, OR analysis showed a significant association of rs2227982 AG with increased risk of TB under the additive model. *PDCD1* is located on chromosome 2, and rs2227982 (+7625 C>T) lies in the fifth exon of *PDCD1*. The rs2227982 base substitution can lead to an exchange of valine for alanine, possibly influencing the activity of PD-1 [29]. In the GTExPortal database, rs2227982 is reported to be associated with *PDCD1* expression (according to expression quantitative trait loci analysis) in subcutaneous and visceral adipose tissues (https://gtexportal.org/home/snp/rs2227982) [30]. In addition, previous studies have indicated associations of *PDCD1* rs2227982 with host susceptibility to hepatitis B infection and risk of *Mycobacterium avium* complex lung disease in women [31, 32]. A significant relationship between serum soluble PD-1 levels and different rs2227982 genotypes also has been reported [32]. PD-1 regulates *Mtb* infection and rs2227982 is associated with PD-1 expression, which may explain the association of rs2227982 with TB risk.

The distribution analyses in the current work revealed a higher frequency in the male population of the rs13170556 TC genotype among patients compared with healthy controls; thus, this genotype may play a predisposing role in TB infection. Human *HAVCR2* is located on chromosome 5, and the SNP rs13170556 lies in the fourth intron of *HAVCR2*. The OR analysis showed a sex-dependent association of the rs13170556 TC genotype with TB risk. Sex-dependent associations of rs4331426 and rs35037722 with TB risk have been reported previously, similar to studies in the Han Taiwanese population [3, 33]. The GTExPortal database indicates an association of rs13170556 with *HAVCR2* expression in testis tissue (https://gtexportal.org/home/snp/rs13170556) [30]. In addition, *HAVCR2* genetic variations have been linked to increased risk of osteoarthritis, possibly because of upregulation of IFN-γ expression by CD4 + T cells [34]. Based on these findings, we hypothesize that rs13170556 may regulate TIM3 and INF-γ expression, in turn influencing susceptibility to TB.

We also found that rs231775 AG and rs231779 CT are associated with reduced risk of TB, similar to research showing that rs231775 AG is a protective factor against TB in a Southern

**Table 6. Haplotype distribution of the two investigated *PDCD1* and *CTLA4* polymorphisms in the overall and sex-stratified populations.**

| Haplotype | | Frequency (%) | TB (n) | Non-TB (n) | *p* value[a] | *p_c* value | Adj. OR (95% CI)[b] | *p* value for OR |
|---|---|---|---|---|---|---|---|---|
| **rs2227982-rs7421861-rs6710479** | | | | | | | | |
| Total | A-A-T (ref.) | 48.4 | 266 | 271 | 0.059 | NS | 1.000 | |
| | G-A-T | 25.7 | 142 | 143 | | | 0.914 (0.669, 1.248) | 0.572 |
| | G-G-C | 15.7 | 91 | 84 | | | 0.973 (0.668, 1.418) | 0.888 |
| | G-A-C | 9.3 | 62 | 41 | | | 1.304 (0.819, 2.075) | 0.263 |
| | G-G-T | 0.6 | 6 | 1 | | | 5.646 (0.586, 54.388) | 0.134 |
| | A-A-C | 0.3 | 3 | 0 | | | ND | ND |
| Men | A-A-T (ref.) | 48.3 | 199 | 216 | 0.069 | NS | 1.000 | |
| | G-A-T | 26.4 | 114 | 113 | | | 1.033 (0.722, 1.478) | 0.860 |
| | G-G-C | 15.2 | 64 | 67 | | | 1.067 (0.687, 1.658) | 0.772 |
| | G-A-C | 9.1 | 42 | 36 | | | 1.035 (0.601, 1.783) | 0.900 |
| | G-G-T | 0.7 | 6 | 0 | | | ND | ND |
| | A-A-C | 0.3 | 3 | 0 | | | ND | ND |
| Women | A-A-T (ref.) | 48.8 | 67 | 55 | 0.059 | NS | 1.000 | |
| | G-A-T | 23.2 | 28 | 30 | | | 0.651 (0.338, 1.251) | 0.198 |
| | G-G-C | 17.6 | 27 | 17 | | | 0.972 (0.461, 2.048) | 0.940 |
| | G-A-C | 10.0 | 20 | 5 | | | 2.876 (0.992, 8.333) | 0.052 |
| | G-G-T | 0.4 | 0 | 1 | | | ND | ND |
| | A-A-C | 0.0 | 0 | 0 | | | ND | ND |
| **rs231775-rs231777-rs231779** | | | | | | | | |
| Total | G-C-T (ref.) | 65.7 | 380 | 349 | 0.618 | NS | 1.000 | |
| | A-C-C | 24.3 | 132 | 138 | | | 0.745 (0.548, 1.012) | 0.060 |
| | A-T-C | 9.9 | 57 | 53 | | | 0.951 (0.616, 1.469) | 0.821 |
| | G-C-C | 0.1 | 1 | 0 | | | ND | ND |
| Men | G-C-T (ref.) | 65.1 | 278 | 282 | 0.995 | NS | 1.000 | |
| | A-C-C | 24.4 | 105 | 105 | | | 0.838 (0.585, 1.200) | 0.335 |
| | A-T-C | 10.5 | 45 | 45 | | | 0.999 (0.613, 1.627) | 0.997 |
| | G-C-C | 0 | 0 | 0 | | | ND | ND |
| Women | G-C-T (ref.) | 67.6 | 102 | 67 | 0.163 | NS | 1.000 | |
| | A-C-C | 24.0 | 27 | 33 | | | 0.488 (0.263, 0.905) | **0.023** |
| | A-T-C | 8.0 | 12 | 8 | | | 0.863 (0.321, 2.323) | 0.771 |
| | G-C-C | 0.4 | 1 | 0 | | | ND | ND |

Abbreviations: Ref., reference haplotype; CI, confidence interval; OR, odds ratio; Pc, the Bonferroni correction of P values.

[a] $\chi^2$ test

[b] Adj. = adjusted for age and sex by logistic regression.

Han Chinese population [27]. In addition, we found that the A-C-C haplotype of the LD block (rs231775-rs231777-rs231779) in *CTLA4* is associated with reduced risk for TB in women. Some previous research has linked this LD block in *CTLA4* with immune-related diseases, such as Graves' disease and Rasmussen syndrome [35, 36]. Human *CTLA4* is located on chromosome 2, and rs231775 (+49A>G) and rs231779 are respectively located in the first exon and intron of the *CTLA4* gene. The rs231775 base substitution at position 17 in exon 1 of *CTLA4* can lead to a change from threonine to alanine [37]. In testis tissue, the association of rs231775 and rs231779 with *CTLA4* expression has been reported in the GTExPortal database (https://gtexportal.org/home/snp/rs231775; https://gtexportal.org/home/snp/rs231779) [30]. In addition, other studies have indicated an association of rs231775 with CTLA-4 expression and an effect on the affinity of CTLA-4 for B7-1 (CD80) in many cancers [38, 39]. *CTLA4* polymorphisms also have been associated with increased secretion of cytokines such as IFN-γ

and IL-2 and an enhanced immune response [40]. Taken together, the findings regarding rs231775 and rs231779 suggest an influence on susceptibility to TB through regulation of CTLA-4 expression and T-lymphocyte response.

The important effects of several host genetic factors on tuberculosis infection have been reported [41]. PD-1, CTLA4, and TIM3 were associated with host immune response against *Mtb* infection [6]. In this study, we indicated the associations of *PDCD1* rs2227982, *HAVCR2* rs13170556, *CTLA4* rs231775, and *CTLA4* rs231779 with TB risk. Though the number of subjects with different TB severity (cavitation and pleural effusion) is not sufficient to evaluate the association of selected SNPs with TB severity in our study, some previous studies have reported the association of these SNPs with clinical outcome of diseases, such as TB, primary biliary cirrhosis, and Hepatitis C [26, 42–44]. In *PDCD1* rs2227982, TB patients with TC or CC genotype have higher rate of tuberculous cavity than those with TT genotype [42]. In previous study of African Population, the *CTLA4* haplotype (rs11571315-rs733618-rs4553808-rs231774-rs231775-rs231777-rs3087243; G-A-A-A-G-C-A) of was associated with cavities in patients with TB [26]. The *CTLA4* haplotype (rs231775-rs231777-rs3087243-rs231725; G-C-G-A) was a protective factor for progression of primary biliary cirrhosis in Japanese patients [43]. Though the association between *HAVCR2* rs13170556 with clinical outcomes of TB is unknown, the protective effects of rs13170556 TC/ CC genotypes with the F protein on the outcomes of HCV infection was reported [44]. In addition, the GTExPortal database indicates an association of *PDCD1* rs2227982, *HAVCR2* rs13170556, *CTLA4* rs231775, and *CTLA4* rs231779 with their gene expression in different tissues. Taken together, these finding may suggest that different genotypes in these SNPs may affect the host immune response against *Mtb* infection and clinical outcome of TB by regulating their gene expression.

The sex-dependent association of *PDCD1*, *CTLA4*, and *HAVCR2* polymorphisms with TB susceptibility were found in our study. Some previous studies also indicated that *PDCD1*, *CTLA4*, and *HAVCR2* polymorphisms were sex-dependently associated with MAC-LD risk, the resolution of hepatitis C virus infection, and the outcomes of HCV infection, respectively [32, 44, 45]. In addition, previous study indicated that the effects of some sex-specific factors on the immune response and women have a higher prevalence of autoimmune diseases compared with men [46]. The difference in function of CD4+CD25+ regulatory T-cells was observed in male and female mice [47]. Taken together, these observations may suggest that the difference in sex-dependent effect of *PDCD1*, *CTLA4*, and *HAVCR2* polymorphisms on immune response against *Mtb* infection. In our study, odds ratio analysis showed the significant associations of heterozygous genotype in *PDCD1* rs2227982, *HAVCR2* rs13170556, *CTLA4* rs231775, and *CTLA4* rs231779 with TB risk. In previous studies, the significant associations of heterozygous genotype of *PDCD1* rs2227982 and *HAVCR2* rs13170556 with the risk for breast cancer and the outcomes of HCV infection were reported, respectively [44, 48]. Though the transcriptional regulatory mechanism of different genotypes in these SNPs did not evaluate in our study, the differences in gene expression between heterozygous and homozygous genotypes of these SNPs in different tissues were found in the GTExPortal database [30]. Based on these findings, we hypothesize that the difference in gene expression between heterozygous and homozygous genotypes of these SNPs may influence immune response against *Mtb* infection. However, our findings still need more researches with large sample sizes to validate. In future, the transcriptional regulatory mechanism of these SNPs and serum levels of soluble PD-1, CTLA4, and TIM3 should be investigated.

Our study has some limitations. First, stratification by sex yielded small group sizes that could have led to an underestimation of significance. Future studies should include a larger number of participants in each group to ensure sufficient statistical power. Second, because

young participants rarely have clinical evidence for the exclusion criteria, it resulted in the significant difference in age of participants between TB and non-TB groups. Third, we did not evaluate some non-genetic factors affecting susceptibility to TB, such as host status (TB contact and latent TB infection) and environment (air pollution and tobacco smoke), and thus cannot exclude the possibility of some influence of these factors in the analyses. In future, to avoid the cofounder effects, OR analysis with adjustment for non-genetic factors of subjects should be performed.

## Conclusions

We identified a significant difference in genotype frequencies of *HAVCR2* rs13170556 between men with and without TB. Our OR analysis showed sex-dependent associations of TB susceptibility and AG heterozygosity at *PDCD1* rs2227982, TC heterozygosity at *HAVCR2* rs13170556, AG heterozygosity at *CTLA4* rs231775, and CT heterozygosity at *CTLA4* rs231779. In addition, the GA/TC combination of *PDCD1* rs2227982/*HAVCR2* rs13170556 was associated with increased TB risk in men, and the A-C-C haplotype at *CTLA4* rs231775-rs231777-rs231779 was associated with reduced risk in women. Overall, *PDCD1*, *CTLA4*, and *HAVCR2* polymorphisms are sex-dependently associated with susceptibility to TB. As the TB statistics of the Taiwan CDC has been showed that men have higher TB prevalence than women [49], our results may help explain the possibility of a difference in TB prevalence between non-TB and TB subjects in men and women. However, our findings still need more researches with large sample sizes in different ethnicities to validate.

## Supporting information

**S1 Table. Characteristics of the selected SNPs.**
(DOCX)

**S2 Table. The sequence of primers and UEP for genotyping assay.**
(DOCX)

**S3 Table. Interaction of genetic variation and sex/age contribute to tuberculosis risk.** Abbreviations: df: degree of freedom. [a]rs231777 TT genotype is only five subjects, none in female subjects and five in male subjects above 60 years old, so the df = 4. [b]rs1036199 CC genotype, rs9313441 AA genotype, and rs919744 GG genotype are not detected in case and control groups, so the df = 3. [c]*p* value was calculated by logistic regression.
(DOCX)

**S4 Table. The differences between groups with and without TB in genotypes and alleles frequencies of selected SNPs and results of odds ratio analysis in non-aged (<65-year-old) participants.** Abbreviations: Ref., reference genotype; CI, confidence interval; OR, odds ratio; Pc, the Bonferroni correction of P values. [a]$\chi^2$ test. [b]Adj. = adjusted for sex by logistic regression.
(DOCX)

**S5 Table. The differences between groups with and without TB in genotypes and alleles frequencies of selected SNPs and results of odds ratio analysis in aged (≥65-year-old) participants.** Abbreviations: Ref., reference genotype; CI, confidence interval; OR, odds ratio; Pc, the Bonferroni correction of P values. [a]$\chi^2$ test. [b]Adj. = adjusted for sex by logistic regression.
(DOCX)

**S6 Table. Haplotype distribution of the two investigated *PDCD1* and *CTLA4* polymorphisms in the age-stratified populations.** Abbreviations: Ref., reference genotype; CI,

confidence interval; OR, odds ratio; Pc, the Bonferroni correction of P values. [a]$\chi^2$ test. [b]Adj. = adjusted for age and sex by logistic regression.

(DOCX)

## Acknowledgments

We thank all the participants in this research for their generous and continued support. Additionally, we thank the Laboratory Medicine Department personnel at Taoyuan General Hospital for their support during the conduct of the study.

## Author Contributions

**Conceptualization:** Lawrence Shih-Hsin Wu, Shih-Wei Lee.

**Data curation:** Chou-Jui Lin, Hsing-Chu Wu, Kuei-Chi Liu.

**Formal analysis:** Lawrence Shih-Hsin Wu, Shih-Wei Lee.

**Funding acquisition:** Shih-Wei Lee.

**Investigation:** Chi-Wei Liu, Lawrence Shih-Hsin Wu.

**Methodology:** Chi-Wei Liu, Chou-Jui Lin.

**Writing – original draft:** Chi-Wei Liu, Lawrence Shih-Hsin Wu.

**Writing – review & editing:** Shih-Wei Lee.

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
