## [Decision Letter · Decision Letter 0]

13 Mar 2024

PONE-D-23-39192Association of tuberculosis risk with genetic polymorphisms of the immune checkpoint genes PD-1, CTLA-4, and TIM3PLOS ONE

Dear Dr. Lee,

Thank you for submitting your manuscript to PLOS ONE. After careful consideration, we feel that it has merit but does not fully meet PLOS ONE’s publication criteria as it currently stands. Therefore, we invite you to submit a revised version of the manuscript that addresses the points raised during the review process.

We look forward to receiving your revised manuscript.

Kind regards,

Afsheen Raza, PhD

Academic Editor

PLOS ONE

Journal Requirements:

5. Please amend your list of authors on the manuscript to ensure that each author is linked to an affiliation. Authors’ affiliations should reflect the institution where the work was done (if authors moved subsequently, you can also list the new affiliation stating “current affiliation:….” as necessary).

**Additional Editor Comments:**

The study is considered important in its field and can provide good background knowledge for scientists working in this field. 

However, the concerns raised by the reviewers is valid and needs to be addressed. 

Reviewers' comments:

Reviewer's Responses to Questions

**Comments to the Author**

1. Is the manuscript technically sound, and do the data support the conclusions?

Reviewer #1: Yes

Reviewer #2: No

2. Has the statistical analysis been performed appropriately and rigorously? 

Reviewer #1: Yes

Reviewer #2: No

3. Have the authors made all data underlying the findings in their manuscript fully available?

Reviewer #1: No

Reviewer #2: Yes

4. Is the manuscript presented in an intelligible fashion and written in standard English?

Reviewer #1: Yes

Reviewer #2: Yes

5. Review Comments to the Author

Reviewer #1: This interesting study investigates the association between SNPs in the immunoregulatory genes with susceptibility to TB in a Taiwan population. The manuscript is well written.

I have some minor suggestions

Please remove the results from introduction section

Please provide the chromosome locations of the studied genes in the introduction

The authors have not mentioned about allele frequencies and its association with TB

The authors have provided separate tables for frequency, and statistical results. It will be difficult for the readers to go back and forth

In one table, they should provide frequencies for total TB and non-TB groups with statistical analysis results including different models. Likewise, a separate table for males and females can be provided

The authors should correct the P values for multiple comparisons

I suggest authors stratify the cases based on disease severity and perform association analysis with SNPs

The age of controls is significantly higher than TB patients. How the authors justify this

In addition to the models analyzed, I suggest analysis by overdominant model since heterozygotes show an association

Reviewer #2: Summary:

The authors tried to investigate the impact of polymorphism s in PDCD1, CTLA4, HAVCR2 on susceptibility for TB infection with total 555 cohort including 285 patients with TB infection. The authors genotyped 11 SNP sites and found that 11 SNPs did not differ significantly in frequency between the non-TB and TB groups. Then, the authors observed that rs2227982, rs13170556, rs231775, and rs231779 were sex-specifically associated with TB risk and concluded that the results may help explain the possibility of a difference in TB prevalence between non-TB and TB subjects in men and women in Taiwan cohort.

Comment

Both genetic susceptibility and impact of exhaustion related gene to TB infection is important topic as prevention medicine and interesting subject in immunology. However, this study has several issues for design, analyzing method and interpretation.

Major

1, TB infection is a known as an opportunistic infection and well investigated the risk factor such as host status and environment for susceptibility. This study design does not care about non genetic factor at all. If authors try to achieve their purpose with current cohort, they should analyze with adjustment factors by generally acceptable cofounders or any other way to avoid the cofounder effects. Currently, the authors do not care a cofounder even the age factor for statistical analyses.

The authors good to summarize the general information about risk factors of TB infection in introduction.

2. There are no description about how the SNPs affect to the susceptibility of TB infection. Some SNPs (or haplotype) must have been reported their functional effect on immune response or clinical outcome in previous reports. The author should try to describe the consistency or inconsistency with reasonable data or explanation.

Why does some SNPs work as sex-specific manner?

How can heterozygous genotype be impactful? Mechanism?

Minor

1. I recommend using the proper symbol for each gene in scientific journal

; for example, PDCD1(italic) gene for PD-1 protein

2. The authors need to provide basic information about SNPs where do they located (intron, exon, or promoter region). The information helps to imagine the biological effect of SNPs.

6. PLOS authors have the option to publish the peer review history of their article (what does this mean?). If published, this will include your full peer review and any attached files.

Reviewer #1: No

Reviewer #2: No

---

## [Author Response · Author response to Decision Letter 0]

18 Apr 2024

Additional Editor Comments: The study is considered important in its field and can provide good background knowledge for scientists working in this field. However, the concerns raised by the reviewers is valid and needs to be addressed.

Reply：Thank you for your kind comment. For your and reviewers’ comments, we modified the Introduction, Results, Discussion, Table 1-6, and S1-S6 Tables. We added two paragraphs in the next to the last paragraph of the Discussion (pages 36-39) in order to discuss the association between selected SNPs and clinical outcome of TB and the sex-dependent association of some selected SNPs with TB risk. In addition, we also modified our manuscript meets the style requirements of PLOS ONE.

Responses to reviewer 1：

Reviewer #1: This interesting study investigates the association between SNPs in the immunoregulatory genes with susceptibility to TB in a Taiwan population. The manuscript is well written.

Reply：Thank you very much for your kind comment. We revised the text in the Introduction, Result, and Discussion and modified the Tables as you suggested. My response to your specific comment is as follows.

In response to the minor comment：

Comment 1: Please remove the results from introduction section.

Reply: Thank you for your comment. As you suggested, we removed the results from introduction section.

Comment 2: Please provide the chromosome locations of the studied genes in the introduction.

Reply: Thank you for your comment. As you suggested, we added the statement about the chromosome locations of the PDCD1, CTLA4, and HAVCR2 genes in the introduction (page 3 and Lines 17-18; page 4 and Lines 10-11; page 5 and Lines 1-2). In addition, we added a new supplementary Table (S1 Table) for the characteristics of selected SNPs.

Comment 3: The authors have not mentioned about allele frequencies and its association with TB.

Reply: Thank you for your comment. As you suggested, we respectively used χ2 test and odds ratio analysis to evaluate the difference in allele frequencies between TB and non-TB groups and the association between allele of SNPs and TB risk. In addition, we added the statement about these results to the text in the Results, Tables 2-4, and S4-S5 Tables.

Comment 4: The authors have provided separate tables for frequency, and statistical results. It will be difficult for the readers to go back and forth.

Reply: Thank you for your comment. As you suggested, to reduce reading difficulties of the readers, we modified Tables 2-5 and S4-S5 Tables and provided genotype frequencies for total TB and non-TB groups with statistical analysis results including different models in one table.

Comment 5: In one table, they should provide frequencies for total TB and non-TB groups with statistical analysis results including different models. Likewise, a separate table for males and females can be provided.

Reply: Thank you for your comment. As you suggested, we added the genotype frequencies for total TB and non-TB groups with statistical analysis results including different models in one table. In addition, we provided separate tables for the statistical analysis results in the overall, male, and female participants.

Comment 6: The authors should correct the P values for multiple comparisons.

Reply: Thank you for your comment. As you suggested, we added the Bonferroni correction of P values in Tables 2-6 and S4-S6 Tables.

Comment 7: I suggest authors stratify the cases based on disease severity and perform association analysis with SNPs.

Reply: Thank you for your valuable comments. As you suggested, we evaluated the percentage of TB severity (cavitation and pleural effusion) in patient with TB. However, the number of subjects with different TB severity (cavity and pleural effusion) is not sufficient to evaluate the association of selected SNPs with TB severity. We added the percentage of TB severity (cavitation and pleural effusion) in Table 1 and the statement about the limitations in our study to the Discussion (page 37).

Comment 8: The age of controls is significantly higher than TB patients. How the authors justify this?

Reply: Thank you for your question. In our study, to avoid the interference of some antiviral and anti-cancer drugs, patients with cancer or other immune-related diseases and viral infections (e.g., hepatitis B, hepatitis C, HIV) were excluded. Because young participants rarely have clinical evidence for the exclusion criteria, it resulted in the age of subjects enrolled in control group was older than TB group. In order to reduce the influence of age of subjects on association of target SNPs with TB susceptibility, we used the logistic regression to simultaneously adjust for age of subjects. In addition, the older people without a history of TB disease seem to be more likely to represent as a control group than young people because they still did not have TB infection. We added the statement about the limitations in our study to the Discussion (page 39) as your suggestion.

Comment 9: In addition to the models analyzed, I suggest analysis by overdominant model since heterozygotes show an association.

Reply: Thank you for your suggestion. According to the definition of various genetic models, the additive model in our study represents overdominant model (wild type homozygous + minor allele homozygous versus heterozygous). As you suggested, we modified the model name in our manuscript.

Responses to reviewer 2：

Reviewer #2: Both genetic susceptibility and impact of exhaustion related gene to TB infection is important topic as prevention medicine and interesting subject in immunology. However, this study has several issues for design, analyzing method and interpretation.

Reply: Thank you very much for your kind comment. We appreciate your providing us the second opportunity to revise the manuscript. We added one new supplementary Table (S1 Table) and two paragraphs in the next to the last paragraph of the Discussion (pages 36-39) as you suggested. My response to your specific comment is as follows.

In response to the major comment：

Comment 1: TB infection is a known as an opportunistic infection and well investigated the risk factor such as host status and environment for susceptibility. This study design does not care about non genetic factor at all. If authors try to achieve their purpose with current cohort, they should analyze with adjustment factors by generally acceptable cofounders or any other way to avoid the cofounder effects. Currently, the authors do not care a cofounder even the age factor for statistical analyses. The authors good to summarize the general information about risk factors of TB infection in introduction.

Reply: Thank you for your comment. In our study, we majorly investigated the association of host genetic polymorphisms with susceptibility to TB. To avoid the interference of non-genetic factors for TB susceptibility, subjects with cancer or other immune-related diseases and viral infections (e.g., hepatitis B, hepatitis C, HIV) were excluded. In addition, the selected participants were majorly enrolled in northern Taiwan, which means our study also excluded the effects of some environmental conditions on susceptibility to TB. 

In this study, we did not evaluate the association of household contacts of TB patients with TB risk. Some TB subjects were household contacts of TB patients. However, it is difficult to clarify whether participants are non-household contacts of TB patients because of consideration of patient confidentiality restraints. In addition, significant differences in age between non-TB and TB groups were found. Because young participants rarely have clinical evidence for the exclusion criteria, it resulted in the age of subjects enrolled in control group was older than TB group. In order to reduce the influence of age of subjects on association of target SNPs with TB susceptibility, we used the logistic regression to simultaneously adjust for age of subjects. We added the statement about the limitations in our study to the Discussion (page 39).

Comment 2: There are no description about how the SNPs affect to the susceptibility of TB infection. Some SNPs (or haplotype) must have been reported their functional effect on immune response or clinical outcome in previous reports. The author should try to describe the consistency or inconsistency with reasonable data or explanation. Why does some SNPs work as sex-specific manner? How can heterozygous genotype be impactful? Mechanism?

Reply: The important effects of several host genetic factors on tuberculosis infection have been reported [Reference 41]. PD-1, CTLA4, and TIM3 were associated with host immune response against Mtb infection [Reference 6]. In this study, we indicated the associations of PDCD1 rs2227982, HAVCR2 rs13170556, CTLA4 rs231775, and CTLA4 rs231779 with TB risk. Though the number of subjects with different TB severity (cavitation and pleural effusion) is not sufficient to evaluate the association of selected SNPs with TB severity in our study, some previous studies have reported the association of these SNPs with clinical outcome of diseases, such as TB, primary biliary cirrhosis, and Hepatitis C [References 26, 42-44]. In PDCD1 rs2227982, TB patients with TC or CC genotype have higher rate of tuberculous cavity than those with TT genotype [Reference 42]. In previous study of African Population, the CTLA4 haplotype (rs11571315-rs733618-rs4553808-rs231774-rs231775-rs231777-rs3087243; G-A-A-A-G-C-A) of was associated with cavities in patients with TB [Reference 26]. The CTLA4 haplotype (rs231775-rs231777-rs3087243-rs231725; G-C-G-A) was a protective factor for progression of primary biliary cirrhosis in Japanese patients [Reference 43]. Though the association between HAVCR2 rs13170556 with clinical outcomes of TB is unknown, the protective effects of rs13170556 TC/CC genotypes with the F protein on the outcomes of HCV infection was reported [Reference 44]. In addition, the GTExPortal database indicates an association of PDCD1 rs2227982, HAVCR2 rs13170556, CTLA4 rs231775, and CTLA4 rs231779 with their gene expression in different tissues. Taken together, these finding may suggest that different genotypes in these SNPs may affect the host immune response against Mtb infection and clinical outcome of TB by regulating their gene expression.

The sex-dependent association of PDCD1, CTLA4, and HAVCR2 polymorphisms with TB susceptibility were found in our study. Some previous studies also indicated that PDCD1, CTLA4, and HAVCR2 polymorphisms were sex-dependently associated with MAC-LD risk, the resolution of hepatitis C virus infection, and the outcomes of HCV infection, respectively [References 32, 44, 45]. In addition, previous study indicated that the effects of some sex-specific factors on the immune response and women have a higher prevalence of autoimmune diseases compared with men [Reference 46]. The difference in function of CD4+CD25+ regulatory T-cells was observed in male and female mice [Reference 47]. Taken together, these observations may suggest that the difference in sex-dependent effect of PDCD1, CTLA4, and HAVCR2 polymorphisms on immune response against Mtb infection. In our study, odds ratio analysis showed the significant associations of heterozygous genotype in PDCD1 rs2227982, HAVCR2 rs13170556, CTLA4 rs231775, and CTLA4 rs231779 with TB risk. In previous studies, the significant associations of heterozygous genotypes of PDCD1 rs2227982 and HAVCR2 rs13170556 with the risk for breast cancer and the outcomes of HCV infection were reported, respectively [References 44, 48]. Though the transcriptional regulatory mechanism of different genotypes in these SNPs did not evaluate in our study, the differences in gene expression between heterozygous and homozygous genotypes of these SNPs in different tissues were found in the GTExPortal database [Reference 30]. Based on these findings, we hypothesize that the difference in gene expression between heterozygous and homozygous genotype of these SNPs may influence immune response against Mtb infection. However, our findings still need more researches with large sample sizes to validate. In future, the transcriptional regulatory mechanism of these SNPs and serum levels of soluble PD-1, CTLA4, and TIM3 should be investigated. We added the statement in the Discussion (page 36-39).

In response to the Minor comment：

Comment 1: I recommend using the proper symbol for each gene in scientific journal; for example, PDCD1(italic) gene for PD-1 protein.

Reply: Thank you for your comment. As you suggested, we used the proper symbol for each gene in the text of our manuscript, Figure legend, and Tables.

Comment 2: The authors need to provide basic information about SNPs where do they located (intron, exon, or promoter region). The information helps to imagine the biological effect of SNPs.

Reply: Thank you for your comment. As you suggested, we added a new supplementary Table (S1 Table) for the characteristics of selected SNPs.

---

## [Editor Report · Decision Letter 1]

25 Apr 2024

Association of tuberculosis risk with genetic polymorphisms of the immune checkpoint genes PDCD1, CTLA-4, and TIM3

PONE-D-23-39192R1

Dear Dr. Shih-Wei Lee

We’re pleased to inform you that your manuscript has been judged scientifically suitable for publication and will be formally accepted for publication once it meets all outstanding technical requirements.

Kind regards,

Afsheen Raza, PhD

Academic Editor

PLOS ONE
---

## [Editor Report · Acceptance letter]

29 Apr 2024

PONE-D-23-39192R1 

PLOS ONE

Dear Dr. Lee, 

I'm pleased to inform you that your manuscript has been deemed suitable for publication in PLOS ONE. Congratulations! Your manuscript is now being handed over to our production team.

Kind regards, 

on behalf of

Dr. Afsheen Raza 

Academic Editor

PLOS ONE